# Educators’ Psychosocial Burdens Due to the COVID-19 Pandemic and Predictive Factors: A Cross-Sectional Survey of the Relationship with Sense of Coherence and Social Capital

**DOI:** 10.3390/ijerph19042134

**Published:** 2022-02-14

**Authors:** Yasue Fukuda, Koji Fukuda

**Affiliations:** 1Faculty of Pharmaceutical Sciences, Suzuka University of Medical Science, Suzuka 513-8670, Mie, Japan; 2Faculty of Political Science and Economics, Waseda University, Tokyo 169-8050, Japan; fukudak@waseda.jp

**Keywords:** COVID-19, educator, sense of coherence, social capital, social and psychological burdens

## Abstract

This study aims to identify the social and psychological burdens placed on educators during the third wave of the COVID-19 pandemic in Japan and to propose an optimal form of support. We investigated educators’ perceptions of psychological and socioeconomic anxieties and burdens, sense of coherence, and social capital using a questionnaire survey of 1000 educators in January 2021. Multivariate regression analyses were conducted to analyze the associations between the variables. Results: Approximately 80% of the respondents considered COVID-19 a formidable, life-threatening illness. Our results revealed that the higher the social capital, the greater the fear of COVID-19, and the higher the sense of coherence, the lower this fear. Conclusions: The anxiety burden of implementing infection prevention was higher than the anxiety burden associated with distance learning. The predictive factors for educators’ perceptions of burden included sense of coherence, gender, and age. Our findings suggest the importance of having the government and educational institutions provide multidimensional assistance that matches educators’ individual characteristics.

## 1. Introduction

The coronavirus disease (COVID-19) pandemic has necessitated the implementation of stringent infection countermeasures. However, equally important is the continuation of educational activities, not only at sites of medicine but also at educational sites that entail numerous group activities [1]. There are concerns among young people in their teens and 20s about infections caused by the variants of the coronavirus, as well as about the spread of household infections from young to old people [2,3]. Since schools conduct countless group activities, there has been considerable concern about the emergence of clusters [4,5,6,7]. Schools in Organisation for Economic Cooperation and Development (OECD) countries were closed for at least 10 weeks [8]. Thus, during the COVID-19 crisis, the educational community made efforts to sustain their educational activities by using the Internet, on-demand, and other modes. Nonetheless, according to Hanushek and Woessmann [9], school closures due to COVID-19 in 2020 catalyzed a 10% reduction in students’ skills, corresponding to a 1.5% reduction in the United States of America’s gross domestic product. This triggered drastic changes in educational methods, including a shift to online classes [8,10,11]. Moreover, educational inequality among students due to differences in family environments [12] and mental health issues were pointed out [13,14,15], which led to growing calls to “liberate” the schools. Not only Japan but also other countries such as Spain and Australia reopened their schools despite not having made sufficient preparations.

On the one hand, the reopening of schools has huge benefits for the students; there are also significant advantages for families, as parents are better able to concentrate on work [16]. However, the reopening of schools is a tradeoff with infection risks, engendering problems such as how to strike a balance between maintaining the quality of education and securing educational opportunities while preventing infections [17,18]. The outbreak of COVID-19 was a turning point for teachers to adopt a variety of measures and programs to guarantee the quality of education, such as using both remote and face-to-face modalities [19,20].

There have been many problems with the Internet-based education provided during the pandemic; these include insufficient communication between the students, as well as between students and teachers [21,22]. As a result of remote education, teachers are forced to cut their rest time, even as they are at home, and send emails to parents and students to answer their individual questions. Furthermore, educators need to spend a lot of time and energy to prevent student infections, while also ensuring the quality of education [8,23]. Policy changes were made in line with the phases of the COVID-19 pandemic; these included transitioning from traditional face-to-face classes to online classes, resuming face-to-face classes from online classes, and using a hybrid modality [10].

During the COVID-19 crisis, anxiety has been increasingly reported among many educators [24]. There are many studies on the mental health of medical professionals, considering their status as essential workers [24,25,26]. The same is true for educators, as they too deal with the COVID-19 pandemic as frontline workers who seek to guarantee continued education [27]. While several studies have been conducted on the mental health of teachers and students [28,29,30,31], the factors that influence educators’ perceptions of increased burden and anxiety, or deteriorating mental health status, have not yet been identified. With regard to the provision of education, fear of COVID-19 and the associated burdens are to be dealt with by the educator—specifically, the financial burden, time consumption, physical and mental health effects of infectious disease countermeasures, and the burden of providing distance education with family members and colleagues.

A sense of coherence (SOC) is regarded as an orientation (in other words, part of an individual’s personality) toward an adaptive nature that enables a person to effectively deal with unfavorable experiences [32,33].

Human health is not solely determined by biological factors. The health status is influenced by individual social backgrounds and problems. These social factors affecting health are called social determinants of health (SDH) [34]. Research on social capital can be traced back to the work of Hanifann [35], who discussed school education and communities. However, this concept was popularized after a series of sociological research studies by Bourdieu [36]. This series was succeeded by sociological research on the relationship between the achievement of youth education and the community by Coleman [37]. Furthermore, Putnam [38] put forth a political science perspective and highlighted the reality regarding the negative impact of reduced social capital as a collective resource for political participation, education, economy, and health. Additionally, social capital acts as a personal resource with characteristics that influence individual education, employment opportunities, health, and as a collective resource within the nation context, it influences social organization. Social capital research has been guided by the major trends of these two characteristics. The study as a personal resource argues that an individual’s income, health status and family status are related to the social network.

Existing research on the effects of social capital as a collective resource focuses on “social factors as an environment,” including social and public health policies. Berkman and Kawachi’s research, which focuses on social capital in relation to health, is also part of the research that relies on the above-mentioned previous research, regarding a personal and collective resources, rather than a unique perspective [39]. Several previous studies have suggested the relationship between social capital and health, and SDH has been shown to affect nearly 60% of individual health [40]. Moreover, a strong association has been indicated between social capital and COVID-19, and a study reports that the existence of social capital leads to an information network and is useful for infection control. However, another study reveals that excessive social capital results in the spread of infectious diseases over time [41].

Fear of COVID-19 has also been reported to impose stress on many essential workers, such as healthcare professionals [24,25], but educators also come in contact with multiple people and groups, including students, and several are infected. This occurrence of illness results in fear and stress; however, these factors have not been fully studied amidst the challenges of educators.

By clarifying the relationship between SOC and everyday perceptions due to a lack of social connectivity or social capital, it can better help to understand the role of educators’ individual characteristics. Studies evaluating the psychosocial burden on educators in the COVID-19 era have recently emerged. Owing to the increased burden on educators, there are concerns about the extent to which burnout syndrome and turnover at educational sites increase teachers’ job burdens. There are also concerns that this increased burden on educators may trigger burnout and turnover [42,43]. Changes in the pandemic situation can also affect the well-being of educators, with the added psychosocial and economic burden, including the risk of spreading COVID-19 infection to educators, their families, and students. Under these circumstances, it is important to investigate how psychological stress develops and accumulates in teachers. Identifying the predictive factors would make it possible to support educational institutions and assist individual teachers.

Therefore, as shown in Figure 1, the purpose of this paper is to elucidate the impact of educators’ burden and educational environment on the use of both online and face-to-face classes according to the continuous spread and change of infectious diseases.

This is the first study to answer the above questions and clarify the national-level situation of all educational institutions. At the same time, we investigated whether individual differences arose in terms of educators’ perceptions of increased burden and anxiety, and if they were apparent, we identified the factors that influenced such differences. The influence on burden and anxiety of educators due to differences in individual attributes would help in designing tailormade support for educators. It can also contribute to quality assurance of education and better human resources management [46].

## 2. Materials and Methods

### 2.1. Study Design and Participants

A cross-sectional survey was conducted online from January 8 to 11, 2021 across educational institutions in Japan, and multivariate analysis was employed on the results of questionnaires obtained from the responses of 1000 employed educators.

A minimum sample size estimate of 385 was derived based on a normal approximation of the binomial distribution with a finite population correction applied (assuming an observed percentage of participants who chose a specific response option of 50%). Based on the statistics of the Ministry of Education, Culture, Sports, Science and Technology, it reflected the proportion of the population size of about one million educators, with a confidence level of 95% and a margin of error of 5%. Based on previous studies, we chose to collect 1000 questionnaires to improve the validity of the survey.

The target participants were men and women who were 20 years of age or older, were members of Japanese educational institutions, and expressed their willingness to participate after receiving an explanation of the research plan on the Internet. Participants who differed from the registered contents, such as age and gender, were excluded from the survey. In addition, to eliminate participants who did not read the five-limb selection question carefully, we set up a trick question asking participants to choose a specified symbol. Those who did not select the specified symbols were excluded. Gender was evenly assigned to each age group, and the questionnaire was distributed to 6000 people who registered their occupations as educators. A total of 1964 people agreed to participate, but we excluded 326 whose survey responses did not match their age, gender, and educational background at the time of enrollment. The number of complete answers was 1175, of which 151 did not answer the trick question properly and were excluded. We also excluded data from 24 participants who recorded significantly shorter response times compared to the others. Finally, the responses of 1000 participants, which were complete and consistent, were used for the analysis.

### 2.2. Measures

The questionnaire items and contents were devised based on previous studies and through discussions among the researchers. The survey included the following sections:

#### 2.2.1. Sociodemographic Characteristics

Distinctions such as demographic characteristics (attributes), including gender and age group, affiliated educational institution (type), education, and fixed-term or full-time employment were noted.

#### 2.2.2. SOC Scale

The Japanese version of the SOC-13 scale was used to measure the psychological stress resistance [47]. It consists of 13 items rated on a seven-point Likert scale and has three subscales: meaningfulness (four items), comprehensibility (five items), and manageability (four items). SOC is used as an index to represent the meaning of situations and diseases, as well as their understanding and manageability. The Cronbach’s reliability coefficient alpha was 0.803.

#### 2.2.3. Health-Related Social Capital

We measured social capital using Saito et al.’s [48] social capital index, which comprises three subscales. The subscales of residential participation (five items) and social cohesion with the regional community (three items) were scored on a four-point Likert scale (1 = strongly agree/yes, 2 = moderately agree/yes, 3 = neither, 4 = disagree/no). Participants who chose strongly or moderately agree/yes were coded 1; others were coded 0. On the mutual support subscale (three items), participants who reported participating in the community at least once a month, having trust, and engaging in mutual help were scored 1. We assigned a score of 0 if participants chose the options of “don’t know” or participated in the community less than once a month. The Cronbach’s reliability coefficient for this scale was 0.693.

#### 2.2.4. Fear of COVID-19

Five items concerning fear of COVID-19 were included based on previous studies [49,50,51,52]. These items were scored using a five-point Likert scale (1 = strongly disagree, 2 = disagree, 3 = neither, 4 = agree, 5 = strongly agree).

Previous surveys on fear and anxiety regarding COVID-19 included, in addition to emotional questions, questions pertaining to symptoms such as “sweaty hands,” “pounding heart,” and “insomnia” [49,50,51,52]. After an internal discussion among the researchers and a pilot survey with the educators, we decided to adopt the insomnia item alone, as it is reported to be caused by fear of COVID-19 [53].

We also changed an item from “It makes me uncomfortable to think about corona” to “Prejudice and discrimination against patients with COVID-19 is emerging.” This is because discrimination on the basis of infection is an important issue given the spread of the infection [54]. Cronbach’s alpha was 0.663.

#### 2.2.5. Psychosocial Burden during the COVID-19 Pandemic in the Educational Environment

After a discussion, the researchers drew up the questionnaire items on the perceptions of psychosocial burden/anxiety experienced by educators concerning education during the COVID-19 pandemic.

Aperribai et al. [28] explored educators’ mental and physical health during the COVID-19 pandemic, while Kim et al. [55] examined the psychological stresses and burdens of educators associated with the reopening of schools, classifying them into five categories. Based on these studies, we included 17 items across the following five areas: (1) anxiety/burden regarding infection and preventive measures at sites of education (four items); (2) anxiety/burden due to the need to cope with the changes in class format (four items); (3) economic anxiety/burden (three items); (4) a perception of increased burden based on shortage of, and reduction in, hours of sleep, rest time, study time, etc. (three items); and (5) issues of social relationships (three items). Responses were evaluated on a five-point Likert scale (1 = strongly disagree, 2 = disagree, 3 = neither, 4 = agree, 5 = strongly agree). Cronbach’s alpha was 0.931.

### 2.3. Ethical Considerations

This study was approved by the Research Ethics Review Board of Waseda University (No 2020-297). Educators were informed of the research plan and asked to check a box to indicate their consent to participate. Those who agreed to participate proceeded to the questions. Participants could withdraw their consent in the middle of an answer and cancel their participation. In such a case, the person’s data would be excluded from the analysis.

### 2.4. Statistical Analysis

The demographic characteristics of interest were as follows: gender, age group, affiliated educational institution, and educational background. Continuous variables were summarized as means and standard deviations. Categorical variables were summarized as frequencies and percentages. The fear related to COVID-19 and the burden on teachers when it becomes widespread is a measure that combines several questions. As in previous studies, each rating scale was treated as a continuous variable and the association between the fear of COVID-19 and predict factor; Based on previous research, on the assumption that the fear of COVID-19, the burden on educators due to COVID-19, and their influential factors are in a linear relationship, a linear regression analysis was performed in this study [56,57]. We performed multivariate linear regression analyses to examine the associations between the outcome variables (SOC, fear of COVID-19, and burden and anxiety regarding education during the COVID-19 pandemic) and the independent variables (social capital and sociodemographic characteristics). Cronbach’s alpha was used for reliability analysis. Data were analyzed using SPSS Statistics version 26 (IBM Corp., Armonk, NY, USA). The significance level was set at 5%.

## 3. Results

### 3.1. Sociodemographic Characteristics

Table 1 shows the participants’ sociodemographic characteristics.

At the time of the survey, the number of infected persons had risen for the third time since the beginning of the pandemic. In response, elementary, middle, and high schools temporarily conducted remote classes. Following a request from the Ministry of Education, Culture, Sports, Science and Technology, these schools reopened in May 2020, and universities were asked to make efforts to increase face-to-face classes after September 2020.

According to data from the Ministry of Education, Culture, Sports, Science and Technology in 2020, the total number of educators in elementary schools, junior high schools, high schools, junior colleges, universities, and graduate schools in Japan was 1,088,212. The proportions of female teachers were 62% in elementary school, 43% in junior high school, 32% in high school, and 25% in college.

We conducted a questionnaire survey considering Japanese educators’ gender ratio, and in terms of the gender ratio of our participants, three-quarters were men. Furthermore, 64% were over 50 years of age. As for affiliated educational institutions, 24.8% worked in elementary schools, 18.3% in middle schools, 29.2% in high schools, 6.6% in junior colleges, and 21.1% in universities.

### 3.2. SOC Scale and Predictive Factors

The average SOC score was 55.61 ± 10.54 (range: 13–91), which is similar to the average SOC of Japanese people [47].

Table 2 shows the results of a multivariate analysis of SOC and the participants’ attributes.

No relationships were observed between SOC and gender, affiliated educational institution, or academic history. However, a relationship was observed between SOC and age group and the status of receiving outpatient treatment (yes or no) (*p* < 0.01). In the comparison of age groups between the 50s and 60s and below 40s, the results indicated SOC to be significantly higher among older adults, such as those in their 50s and 60s (Tukey’s test, *p* < 0.01).

### 3.3. Social Capital in the Region

The average social capital score was 4.09 ± 2.00. No relationships were observed between social capital and age group, gender, affiliated educational institution, academic history, or form of employment (Tukey’s test, *p* > 0.05).

### 3.4. Anxiety/Fear of COVID-19 and Predictive Factors

Table 3 shows the statistical results for the participants’ descriptions of their fear of COVID-19.

The scores showing insomnia to be caused by COVID-19 were low (1.95 ± 1.06). However, the scores for “COVID-19 is a scary disease” and “My life is being threatened by COVID-19” were both high (4.26 ± 0.91 and 4.08 ± 0.95, respectively). Thus, discrimination against people with COVID-19 and fear of social exclusion (3.85 ± 1.05) were more common than experiencing anxiety and nervousness upon encountering news about COVID-19 (3.25 ± 1.17). The average of the sum total was 17.94 ± 3.37 (range: 5–25).

Table 4 shows the results of a multivariate analysis of fear/anxiety regarding COVID-19 and the participants’ sociodemographic characteristics, SOC, and social capital.

A negative correlation was observed, in which the higher the SOC scores, the lower the fear of COVID-19. The results for men and women (17.04 ± 3.34 and 18.42 ± 3.31, respectively) suggested that women had a significantly greater fear of the disease (*p* < 0.01). Furthermore, there was a positive correlation between social capital and fear of COVID-19, and a negative correlation between fear of COVID-19 and SOC and academic degree.

### 3.5. Perception of Psychological, Social, and Economic Burden and Anxiety at Sites of Education

Table 5 shows the statistical results of the participants’ perceptions of increased burden/anxiety at the sites of education. Psychological, social, and financial burdens and anxieties in educational activities were significantly higher for infectious disease-related anxiety than for distance learning (*p* < 0.01). In addition, anxiety and concern about a lack of connections with patients, families, and students were significantly higher than the sense of burden in terms of time and economy (*p* < 0.01) (Tukey’s test, *t*-test).

The average of the sum total was 61.11 ± 12.32 (range: 17–85).

Table 6 shows the results of multivariate analysis of the perceptions of psychosocial anxiety or burden at the sites of education and the participants’ fear of COVID-19, sociodemographic characteristics, SOC, and social capital.

The perceptions of psychological, social, and economic burden and anxiety at sites of education correlated positively with social capital and anxiety about COVID-19. Furthermore, the perceptions of psychological, social, and economic burden and anxiety at sites of education correlated negatively with SOC and age group: the higher the SOC and age group, the lower the perception of psychosocial anxiety/burden.

## 4. Discussion

This study was carried out while a newly emerging infectious disease, COVID-19, had broken out and was not being adequately controlled, even after more than a year, with educators being forced to continue carrying out both infection countermeasures and educational activities. Previous studies that investigated the psychosocial issues faced by educators and students were conducted during the early stages of the COVID-19 pandemic or during school lockdown [13,14,15,16,21,28,29,30,31,53,55,58,59,60]. Our study shows the subsequent stages during spread and that they converged repeatedly, prolonging the period of infectious diseases. This study is unique and provides insight into the psychological burden of educators. Based on our research findings, we aim to support the long-term exhaustion within the educational field, while ensuring the quality of education in the future.

In this context, we clarified the socio-psychological and economic burden that COVID-19 placed on educators. Educators’ burden and anxiety arose from the following five factors: (1) economic burden, (2) time constraints, (3) infection control measures, (4) distance education technostress, and (5) maintaining social ties. Educators’ burden and anxiety associated with the implementation of education were associated with fear of COVID-19, age, gender, SOC, and social capital.

First, our study suggested that the participants were compelled to continue working as educators while harboring anxieties about education-related concerns such as the spread of infection among students, the impact on their nurturing of sociability, and concerns about isolation. Regarding the fear of an infectious disease and the burden on educators, 80% of the educators considered COVID-19 to be a formidable disease. Gender differences were recognized, with women being under greater stress than men. This is consistent with a previous study on the fear of COVID-19 among Mexican high school teachers [24]. However, there are few studies on educational stress and burden due to COVID-19. During the ongoing COVID-19 crisis, educators have been requested to conduct classes using both online and face-to-face modes. COVID-19 not only puts people’s health at risk but also contributes to mental health and economic problems such as employment instability and burden and anxiety at the sites of education [30]. Educators have been required to engage more frequently with the use of remote education techniques and the Internet, in addition to interaction with students and their parents. As a result, educators have less time for research and preparation of teaching materials. Their work time and workload have increased, and especially teachers with chronic illnesses and those living with family members at high infection risk experience heavy material, economic, and mental burdens associated with the need to apply anti-infection measures. There are several studies on the psychological effects of COVID-19 on educators’ awareness of the transition to online classes and technostress [29,59,61]. According to Worth and Van den Brande [62], professionals in education are under more stress than those in other professions. In addition, Perryman and Calvert [63] pointed out that teachers are under increasing pressure from school accountability systems. A study, conducted in the University of Jordan, regarding psychological stress during the COVID-19 pandemic found that over half the faculty members were reportedly experiencing stress due to COVID-19 [59]. According to Aperribai et al.’s [28] study on educators’ mental health during the lockdown, the observed problems included higher working hours, technical stresses in online classes, inability to directly teach students, and problems related to social relationships in their families and workplaces. However, there are few empirical studies on the impact on educators of the transition from online to face-to-face classes. In addition, there are few reports on the impact of both remote and face-to-face classes on educators and the educational environment. Thus far, most studies have focused on the evaluation of psychological stress experienced by educators using the Perceived Stress Scale and General Health Questionnaire, among others, and have pointed out the need for psychological support [28,64,65]. A study on COVID-19 by Al-Sabba et al. [65] examined the feeling of well-being among university students and faculty members of the Department of Health Psychology at two universities amidst school closures. The morbidity of COVID-19 reveals that symptoms such as insomnia and anxiety occur, negative emotions and happiness decrease. However, improved communication enhances happiness, and happiness is not related to educational background or gender. This finding is also consistent with our research results. According to a study by Eadie et al. [66], a strong positive correlation exists between the educator’s and students’ well-being among students in their early childhood. This finding can also be generalized to the relationship between educators and students in other age groups, so it is necessary to take measures to reduce the burden on educators. Since our research has revealed stress as an influential factor, we believe that the relationship between educators and students can be improved by providing support to eliminate this factor.

A study by Spadafora et al. [67] examined the relationship between kindergarten teachers’ mental health and their care for children and adolescents and indicated that family responsibilities affect their mental health. Since our study did not examine the family composition, future research should examine the effects of family composition and family relationships.

Second, our study found that the higher the SOC score, the lower the burden of education implementation due to commitment. In line with our results, a survey conducted in Nagasaki, Japan, which examined the mental health of high school teachers (measured via the General Health Questionnaire 12) and SOC, found that the higher a person’s degree of satisfaction with work, sense of self-control, and SOC, the better the mental health status [68]. It has been pointed out that SOC—a measure of psychological stress-related resilience—and social capital are related to health. To support and reinforce educational activities, educational institutions, the government, and administrative bodies need to provide diverse forms of assistance to alleviate educators’ mental and physical stress and reduce their anxiety levels [10,20]. A correlation between psychological stress-related resilience (SOC) and age has also been previously reported [69], supporting the current results. In addition, our study revealed that the ongoing COVID-19 pandemic is increasing educators’ anxieties and burdens regarding infection prevention and disease management. According to Lizana et al.’s [70] study on the quality of life of educators during the COVID-19 pandemic, the quality of life of women under 45 was significantly reduced compared to pre-pandemic levels. Our study indicates that the burden and anxiety of educators are related to age and gender. According to Akour et al.’s [59] study on university faculty members, more than half experienced the fear of being infected with COVID-19 and exhibited social resistivity, which is consistent with our study. According to Penado Abilleira et al. [61], the technostress of older educators transitioning from face-to-face to remote classes is higher than that of younger educators.

Third, our study showed that the higher the social capital, the greater the burden of education during the COVID-19 pandemic. Social capital leads to a teacher’s professional identity [71]. Moreover, it can be considered that the higher the professional awareness, the higher the social capital. Our study reveals a relationship between social capital and the high burden of providing education related to factors such as infectious disease control, concerns with family members, and colleagues. More enthusiastic teachers with higher social capital and higher professionalism may be able to deal with various psychological, time, and physical burdens. Regarding the effects of social capital on health, Bai et al. [72] investigated the link between social capital and social distancing among American individuals during the COVID-19 crisis and reported that the higher the social capital, the more socially distant their behavior. This finding can be interpreted as taking preventive action in society to maintain social capital. Our study suggests that during the chronic crisis brought about by the COVID-19 pandemic, sincere educators with higher social capital are taking measures against infection and ensuring the quality of education, with a feeling of burden. In addition, a large study by Bartscher et al. [41] on social capital in the European region pointed out that there is an awareness that it is a citizen’s duty to voluntarily ensure social distance. The COVID-19 pandemic has accelerated the spread of online classes and digitization of educational activities, with the educational industry being tasked with playing a role in offering new educational platforms. However, it is becoming increasingly clear that some types of learning cannot be replaced with online classes, and that it is difficult to provide field training remotely in medical education, long-term care, and the welfare sector [73].

The knowledge and findings obtained from this study make it possible to implement appropriate and specific support measures for educators, which the government and educational institutions are requested to tackle, and thereby improve the quality of education. Concerning educators’ fear of COVID-19, our findings suggest that the higher their social capital, the greater their perception of increased burden at the sites of education. This is possibly because people with high social capital who carry out social activities other than at their place of work, such as volunteers, could no longer maintain social connections as before, or they may avoid making interpersonal contacts or exchanges during the COVID-19 pandemic; therefore, they had a greater perception of increased burden and anxiety. Offering professional assistance to teachers on learning management systems or digital technologies and assessing the influence of such technologies on educators are essential to sustain the delivery of education and guarantee its quality in the context of the COVID-19 pandemic. It is, thus, necessary to consider what kind of support should actually be provided and to whom, and to provide tailored assistance to prevent mental health problems and offer educational support. Shedding light on the educators’ views on their burdens and anxiety related to implementing education will make it possible for the government and educational institutions to offer educators realistic and concrete support [74,75,76,77,78].

### Limitations

This study has several limitations. First, because we used an Internet-based survey, it is possible that a large number of participants belonged to the Internet-using population, and that the perception of increased burden when teaching remote classes may be underestimated. Second, although we distributed the questionnaire with the intent of eliminating gender differences, the distribution of participants may indicate a gender bias. Third, we did not enquire about individual circumstances using free description. Further studies, using qualitative methods such as interview surveys, on educators’ long-term conditions during and after the COVID-19 pandemic are needed. Fourth, the research period of our study was limited; therefore, future studies should consider the long-term impact of COVID-19 on educators. Fifth, our study period coincided with the development of COVID-19 vaccines globally; however, at the time of the study, the access to vaccines in Japan was limited. Therefore, further research is required to observe the changes amidst the vaccination rates and COVID-19 mutational conditions.

## 5. Conclusions

Our study identified the factors that induce stress and anxiety, such as economic and environmental factors, sociability, and isolation, as well as factors that cause individual differences. To secure opportunities for education in the context of the COVID-19 pandemic and to guarantee educational quality, it is essential to evaluate the pandemic’s influence on educators. Educational institutions can provide better support to educators, made possible by shedding light on the mental health of not only students but also teachers as well as their perception of increased labor burden and factors that cause anxiety. Our study revealed that although the pandemic has not caused anxieties and fears that are serious enough to cause insomnia, it has increased the frequency at which the teachers use remote education techniques, cope with using the Internet, and deal with the students and their parents/guardians. Their labor time and financial burden have increased along with the fear of COVID-19. Moreover, educators who have an underlying disease, who are of advancing age, and who live with a family member at high infection risk bear a substantially heavy burden—material, economic, and mental—due to the need to implement infection countermeasures at the sites of education. It was also revealed that they are compelled to engage in educational activities while harboring anxieties in other educational areas, such as nurturing students’ sociability as well as concerns about isolation of the self, colleagues, and family. Furthermore, since a strong correlation was found for factors such as SOC and age as predictive factors of teachers’ perception of increased burden, it is important for educational institutions to provide educators with multidimensional support that is tailored to their individual characteristics.

## Figures and Tables

**Figure 1 ijerph-19-02134-f001:**
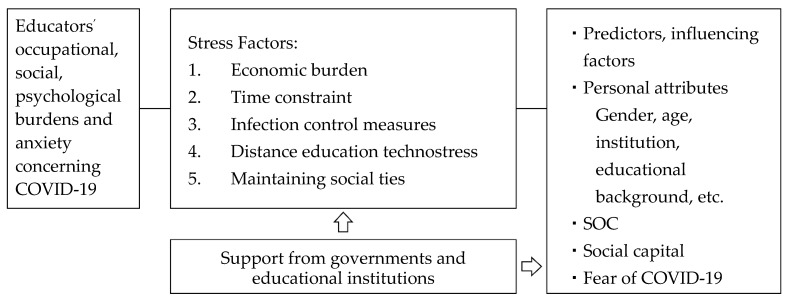
The burden of educators and the impact on the educational environment during COVID-19. (SOC = sense of coherence). We formulated this figure of analytical approach to the social and psychological burden imposed by COVID-19 on educators, by referring to the method of NIOSH approach to Job Stress [44,45].

**Table 1 ijerph-19-02134-t001:** Participants’ sociodemographic characteristics.

		Frequency (*n*)	%
Gender	Men	769	76.9
	Women	231	23.1
Age group	20s	29	2.9
	30s	111	11.1
	40s	206	20.6
	50s	408	40.8
	60s	246	24.6
Academic history	University graduate	651	65.1
	Graduate school graduate	314	31.4
	Other	35	3.5
Degree	B.A.	651	65.1
	M.A.	163	16.3
	Ph.D.	151	15.1
	None	35	3.5
Form of employment	Full-time	811	81.1
	Fixed term	65	6.5
	Part-time	124	12.4
Affiliated educational institution	Elementary school	248	24.8
	Middle school	183	18.3
	Senior high school	292	29.2
	Technical college; junior college	66	6.6
	University	211	21.1
Receiving outpatient treatment: Yes/No	Yes	430	43
	No	570	57

**Table 2 ijerph-19-02134-t002:** Factors influencing sense of coherence: linear multivariate regression analysis.

	Non-Standardized Coefficient		Standardized Coefficient	*t*-Value	Significant Probability	95% Confidence Interval	
	B	SD	Beta			Lower Limit	Upper Limit
(Constant)	45.439	2.892		15.712	0.000	39.763	51.114
Gender	−0.560	0.857	−0.022	−0.653	0.514	−2.241	1.122
Age group	1.829	0.362	0.179	5.050	0.000	1.119	2.540
Affiliated educational institution	0.317	0.320	0.043	0.993	0.321	−0.310	0.944
Form of employment	0.259	0.521	0.017	0.497	0.620	−0.764	1.281
Academic history	−3.342	1.890	−0.148	−1.768	0.077	−7.052	0.368
Degree	1.699	1.263	0.120	1.345	0.179	−0.780	4.178
Receiving outpatient treatment: Y/N	1.875	0.705	0.088	2.662	0.008	0.493	3.258

Note: SD = standard deviation; adjusted R^2^ = 0.0.37

**Table 3 ijerph-19-02134-t003:** Educators’ responses regarding fear of COVID-19.

	Strongly Agree		Agree		Neither		Disagree		Strongly Disagree		Mean	SD
	*n*	%	*n*	%	*n*	%	*n*	%	*n*	%		
COVID-19 is a scary disease	483	48.3	368	36.8	99	9.9	26	2.6	24	2.4	4.26	0.914
My life is being threatened by COVID-19	373	37	428	43	128	13	45	5	26	2.6	4.08	0.954
I cannot sleep at night when I think about COVID-19	21	2	69	7	207	21	242	24	461	46.1	1.95	1.064
Prejudice and discrimination against patients with COVID-19 is emerging	266	27	449	45	178	18	53	5	54	5.4	3.82	1.054
News about COVID-19 sometimes makes me anxious and nervous	126	12.6	361	36.1	257	25.7	152	15.2	104	10.4	3.25	1.171

Note: SD = standard deviation; evaluated on a five-point Likert scale from “1. Does not apply” to “5. Applies”.

**Table 4 ijerph-19-02134-t004:** Fear/anxiety regarding COVID-19 and influencing factors (linear multivariate regression analysis).

	Non-Standardized Coefficient		Standardized Coefficient	*t*-Value	*p*-Value	95% Confidence Interval	
	B	SD	Beta			Lower Limit	Upper Limit
(Constant)	18.493	0.925		19.992	0.000	16.678	20.308
Gender	1.335	0.264	0.165	5.060	0.000	0.817	1.852
Age group	0.174	0.109	0.053	1.591	0.112	−0.041	0.388
Workplace	−0.174	0.098	−0.074	−1.769	0.077	−0.366	0.019
Form of employment	−0.017	0.161	−0.003	−0.107	0.915	−0.333	0.299
Degree	−0.817	0.389	−0.182	−2.100	0.036	−1.581	−0.053
Sense of coherence	−0.075	0.010	−0.237	−7.482	0.000	−0.095	−0.056
Social capital	0.153	0.053	0.091	2.894	0.004	0.049	0.256

Note: SD = standard deviation; dependent variable: anxiety about COVID-19; adjusted R^2^ = 0.043.

**Table 5 ijerph-19-02134-t005:** Burden/anxiety at educational sites due to COVID-19.

Environment and Technologies (Changes in Class Format)	Mean	SD	Infection Countermeasures	Mean	SD
Remote technology	3.72	1.124	Anxiety of pupils and infected persons	4	0.922
Internet environment	3.59	1.115	Getting the self, colleagues, and family infected	4.01	0.925
Support setup	3.75	1.065	News about cluster occurrence	3.85	0.97
Increase in dealing with parents and students	3.74	1.065	Tediousness of infection countermeasures	3.89	0.977
	3.7	1.09225		3.9375	0.9485
Total	14.7950	3.81570	Total	15.7510	3.31300
Time			Social relationships		
Shortage of rest time	3.35	1.159	Nurturing of students’ sociability	3.8	0.907
Lack of sleep	3.18	1.151	Students and faculty becoming isolated	3.58	1.014
Reduction in research/study time	3.22	1.113	The self and family becoming isolated	3.3	1.071
	3.25	1.141		3.56	0.9973333
Total	9.7470	3.10300	Total	10.6790	2.61205
Financial/economic aspects					
Employment, economy, anxiety	3.48	1.079			
Economic costs of providing/setting up PCs	3.44	1.081			
Insufficient research fees	3.22	1.076			
	3.38	1.0786667			
Total	10.1330	2.74533			

Note. SD = standard deviation; PC = personal computer.

**Table 6 ijerph-19-02134-t006:** Burden/anxiety at educational sites due to COVID-19 and influencing factors: Linear multivariate regression analysis.

Coefficient ^a^							
	Non-Standardized Coefficient		Standardized Coefficient	*t*-Value	*p*-Value	95% Confidence Interval	
	B	SD	Beta			Lower Limit	Upper Limit
(Constant)	48.768	3.589		13.589	0.000	41.726	55.811
Gender	0.599	0.871	0.020	0.688	0.491	−1.109	2.308
Age group	−0.742	0.356	−0.062	−2.084	0.037	−1.440	−0.043
Affiliated educational institution	−0.317	0.320	−0.037	−0.989	0.323	−0.946	0.312
Form of employment	−0.922	0.524	−0.051	−1.758	0.079	−1.951	0.107
Degree	−0.545	1.271	−0.033	−0.428	0.668	−3.039	1.950
Social capital	0.673	0.173	0.109	3.896	0.000	0.334	1.012
SOC	−0.231	0.034	−0.198	−6.848	0.000	−0.297	−0.165
Anxiety about COVID-19	1.559	0.105	0.423	14.791	0.000	1.352	1.765

Note. ^a^ Dependent variable: burden of, and anxiety about, education; SD = standard deviation; SOC = sense of coherence.

## Data Availability

The datasets generated and/or analyzed during the current study are available from the corresponding author on reasonable request.

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
