# Peer review of "Educators’ Psychosocial Burdens Due to the COVID-19 Pandemic and Predictive Factors: A Cross-Sectional Survey of the Relationship with Sense of Coherence and Social Capital"

_ijerph, 2022, doi:10.3390/ijerph19042134_

Round 1
Reviewer 1 Report
The research study reported in the manuscript focused on a less researched topic, namely, the factors influencing the psychosocial burdens and anxiety of educators during the COVID-19 pandemic. The manuscript is well-organized and free from inconsistencies in terms of presentation and language use. The research study also had rigorous sampling procedures, and results are presented systematically.
It is claimed in the manuscript that “while several studies have been conducted on the mental health of 69 teachers and students [28–31], the factors that influence educators’ perceptions of increased burden and anxiety, or deteriorating mental health status, have not yet been identified” (p. 2, line 70 – 71). I performed a keyword search using "pandemic", "factors", "burdens", and "educators", and the following entries which explored the factors contributing to educators’ psychosocial burdens were found:
Al-Sabba, S., Darwish, A., Fares, N., Barnes, J. & Almomani, J. A. (2021). Biopsychosocial factors linked with overall well-being of students and educators during the COVID-19 pandemic. Cogent Psychology, 8(1). DOI: 10.1080/23311908.2021.1875550
Eadie, P., Levickis, P., Murray, L. et al. (2021). Early childhood educators’ wellbeing during the COVID-19 pandemic. Early Childhood Education Journal, 49, 903–913. DOI: 10643-021-01203-3
Spadafora, N., Reid-Westoby, C., Pottruff, M., & Janus, M. (2021). Family responsibilities and mental health of kindergarten educators during the first COVID-19 pandemic lockdown in Ontario, Canada. https://doi.org/10.1101/2021.05.11.21257057.
There is a need for revising the relevant statements in the manuscript, for example, by adding information on how the research study reported in the manuscript made advancements on the above studies and its uniqueness.
It is not entirely clear why the factor “sense of coherence” (among other factors such as attitudes to life and locus of control) was selected as a factor. Was it because past research has indicated its significance in contributing to anxiety or other similar negative emotions? The literature given in the discussion section (i.e. reference entries 52 – 55) should be included in the introductory section. Similarly, was the construction of the model given in Figure 1 based on any theoretical framework or past research evidence? These important issues need to be addressed.
The questionnaire did not cover all the factors included in Figure 1. There is a need for explaining the discrepancies. Among the factors included in the questionnaire, only sense of coherence was introduced in the introductory section. The significance of “health-related social capital” (2.2.3) and “fear of COVID-19’ should be introduced in the introductory section before they are presented in the methodology section. Again, the relevant reference entries (56 – 59) should be added to the introductory section. An alternative is the add a separate literature review section after the introduction.
Author Response
Thank you for kind advice. Please fined the attached file that provide a point-by-point response to the your useful comments.

Reviewer 2 Report
The only aspect I would probably contextualize a bit better is the construct of "social capital". In the title and in the abstract is called only "social capital" whereas in the text the reader discovers that it is a "health-related social capital".
Given the relevance of the concept of "social capital" in Western Education literature (from Bourdieu on, passing through Putnam and many others), it is absolutely necessary to specify it in the title and in the abstract that this is a different kind of "social capital".
Author Response

(The authors gave the same response as above.)

Reviewer 3 Report
This is an interesting topic. However, the authors need to provide major revisions.
1). Why wasn't an ordered probit regression used as the dependent variable is a cumulative likert score ? Therefore, a strong justification needs to be provided as to why these authors used linear regression. I am very skeptical that there is one such explanation.
2). What time frame were the respondents using -- the first surge, second surge, etc. In other words, specific dates would be helpful. For example, since the pandemic is worldwide, could educators' answers be influenced by other countries' experience.
3). Specifics regarding vaccination rates among teachers. And it appears that medical school educators would be more scientific savvy compared to elementary school teachers. Therefore, levels of teaching institution appear to be a proxy variable and should be explained as such.
Hence, the results, in my opinion, are not based on sound methodological approaches and require a major revision.
Author Response

(The authors gave the same response as above.)

Round 2
Reviewer 1 Report
The issues raised in my previous review have been adequately addressed.
Author Response
Response to Reviewer 1
Comments and Suggestions for Authors
The issues raised in my previous review have been adequately addressed.
Response: Thank you for your valuable comments. They helped us improve the quality of the manuscript. 

Reviewer 3 Report
The authors' responses were not satisfactory -- they provide some obscure citations, but did not, in their own words fully discuss the choice and justification of the linear model chosen.
The time frame appears to be too short, and the explanation appears to be due to expediency with little support for the general use for the results.
Author Response
Response to Reviewer 3
The authors' responses were not satisfactory -- they provide some obscure citations, but did not, in their own words fully discuss the choice and justification of the linear model chosen.
Based on the previous research, the analysis was performed on the assumption that the fear of Covid 19 and the burden on the educator and their influential factors are in a linear relationship.
The time frame appears to be too short, and the explanation appears to be due to expediency with little support for the general use for the results
Response:
Thank you for the suggestion. While we agree that, in general, linear regression is considered inappropriate and that the Likert scale is used; this is because linear regression requires the objective variable to be set as a continuous variable. However, linear regression analysis is possible if the ordinal scale is considered to be evenly spaced.
The fear related to COVID-19 and the burden on teachers when it becomes widespread is a measure that combines several questions. As in previous studies, each rating scale was treated as a continuous variable and a linear regression analysis was performed [56,57]. Based on previous research, the analysis was performed based on the assumption that the fear related to COVID-19, the burden due to COVID-19 on educators, and their influential factors are in a linear relationship. 
On the other hand, the probit analysis that you had recommended is a method of observing the influence of explanatory variables using the reaction probability as the objective variable,
There were little similar studies (regarding the fear related to COVID-19) in which probit analysis was performed.
Hence, we consulted with statistical experts who confirmed that the explanatory variables are linear and can be an appropriate analytical method. Therefore, assuming an objective variable, a linear regression analysis was conducted in this study.
We included the following sentences in our manuscript:
The fear related to COVID-19 and the burden on teachers when it becomes widespread is a measure that combines several questions. As in previous studies, each rating scale was treated as a continuous variable and the association between fear of COVID-19 and predict factor a linear regression analysis was performed [56,57]. Based on previous research and the advice of the statistical experts whom we consulted, the analysis was performed based on the assumption that the fear of COVID-19, the burden on educators due to COVID-19, and their influential factors are in a linear relationship. Additionally, since the explanatory variables are linear it is considered to be appropriate. Hence, assuming an objective variable, a linear regression analysis was conducted in this study.
Fourth, the research period of our study has a limited; therefore, future studies should consider the long-term impact of COVID-19 on educators.
